# Genome-Wide Identification and Analysis of the EIN3/EIL Transcription Factor Gene Family in Doubled Haploid (DH) Poplar

**DOI:** 10.3390/ijms25074116

**Published:** 2024-04-08

**Authors:** Caixia Liu, Erqin Fan, Yuhang Liu, Meng Wang, Qiuyu Wang, Sui Wang, Su Chen, Chuanping Yang, Xiangling You, Guanzheng Qu

**Affiliations:** 1College of Life Science, Northeast Forestry University, Harbin 150040, China; liucaixia2020@outlook.com (C.L.); youxiangling@nefu.edu.cn (X.Y.); 2State Key Laboratory of Tree Genetics and Breeding, Northeast Forestry University, Harbin 150040, China; fanerqin2012@hotmail.com (E.F.); m13091438838@163.com (Y.L.); wangmengnefu@163.com (M.W.); wqy19821220@163.com (Q.W.); chensu@nefu.edu.cn (S.C.); yangcp@nefu.edu.cn (C.Y.); 3State Key Laboratory of Tree Genetics and Breeding, Key Laboratory of Tree Breeding and Cultivation of National Forestry and Grassland Administration, National Innovation Alliance of Catalpa Bungei, Research Institute of Forestry, Chinese Academy of Forestry, Beijing 100091, China; 4Key Laboratory of Soybean Biology in Chinese Ministry of Education, Northeast Agricultural University, Harbin 150030, China; wangsui.ws@163.com

**Keywords:** doubled haploid (DH) poplar, EIN3/EIL, identification, expression analysis

## Abstract

Ethylene (ET) is an important phytohormone that regulates plant growth, development and stress responses. The ethylene-insensitive3/ethylene-insensitive3-like (EIN3/EIL) transcription factor family, as a key regulator of the ET signal transduction pathway, plays an important role in regulating the expression of ET-responsive genes. Although studies of EIN3/EIL family members have been completed in many species, their role in doubled haploid (DH) poplar derived from another culture of diploid *Populus simonii* × *P. nigra* (donor tree, DT) remains ambiguous. In this study, a total of seven EIN3/EIL gene family members in the DH poplar genome were identified. Basic physical and chemical property analyses of these genes were performed, and these proteins were predicted to be localized to the nucleus. According to the phylogenetic relationship, EIN3/EIL genes were divided into two groups, and the genes in the same group had a similar gene structure and conserved motifs. The expression patterns of EIN3/EIL genes in the apical buds of different DH poplar plants were analyzed based on transcriptome data. At the same time, the expression patterns of *PsnEIL1*, *PsnEIN3*, *PsnEIL4* and *PsnEIL5* genes in different tissues of different DH plants were detected via RT-qPCR, including the apical buds, young leaves, functional leaves, xylem, cambium and roots. The findings presented above indicate notable variations in the expression levels of *PsnEIL* genes across various tissues of distinct DH plants. Finally, the *PsnEIL1* gene was overexpressed in DT, and the transgenic plants showed a dwarf phenotype, indicating that the *PsnEIL1* gene was involved in regulating the growth and development of poplar. In this study, the EIN3/EIL gene family of DH poplar was analyzed and functionally characterized, which provides a theoretical basis for the future exploration of the EIN3/EIL gene function.

## 1. Introduction

Phytohormones (plant hormones) are naturally-occurring organic substances that act independently or in conjunction to influence physiological processes at low concentrations [1,2]. Ethylene (ET), a gaseous phytohormone with a simple C2H4 structure, facilitates communication between plants and their environment [3,4]. It serves various functions, including regulating leaf development, shaping plant growth, inducing senescence, promoting flower development and fruit ripening, stimulating germination and responding to environmental stresses, both abiotic and biotic [5,6,7,8,9,10,11,12]. At the same time, ET has also been shown to induce typical morphological changes in dark-grown seedlings known as the triple response, including the inhibition of hypocotyl and root elongation, radial swelling of hypocotyl and exaggeration of the apical hook [13]. There are five ET receptors localized on the endoplasmic reticulum membrane in *Arabidopsis*, including ethylene response 1/2 (ETR1/2), ethylene response sensor 1/2 (ERS1/2) and ethylene insensitive 4 (EIN4). In an environment with an extremely low ET concentration, the receptor binds to the N-terminus of the constitutive triplet reaction 1 (CTR1) and phosphorylates the C-terminus of EIN2, making it easier to be degraded, thereby inhibiting ET signaling. EBF1/2, as F-box proteins, recognizes and marks EIN3/EIL as the target of the ubiquitination enzyme, but is not directly involved in the degradation process. Subsequently, these EIN3/EILs labeled in the nucleus are degraded through the ubiquitin-proteasome pathway. At the same time, the transcription factor EIN3/EIL is degraded by EBF1/2-mediated ubiquitinase in the nucleus, and the ET signaling pathway is completely inhibited. On the contrary, in an environment with a high concentration of ET, ET molecules bind to receptors, inactivate CTR1 and dephosphorylate EIN2, and its C-terminus is cleaved and transported to the nucleus, where it binds to EIN3/EIL and increases its expression abundance, regulating the expression of related genes by binding primary ethylene response element (PERE) in response to environmental changes [14,15,16]. Therefore, the ethylene-insensitive3/ethylene-insensitive3-like (EIN3/EIL) transcription factor family is a key regulator involved in the ET signal transduction pathway in plants.

The EIN3/EIL gene encodes a nuclear localization protein in higher plants, characterized by a conserved N-terminal DNA-binding domain and a distinctive folding structure [16,17]. There are six genes encoding the members of this family in *Arabidopsis* (EIN3 and EIL1/2/3/4/5) [16,18]. EIN3 and EIL1 exhibit the closest homologous relationship. The overexpression of *EIL1* in *ein3* mutants can complement the loss of *ein3* functions and activate the ET signaling pathway [16]. In the *ebf1* and *ebf2* double mutants, the loss of *EIN3* functions can effectively inhibit the growth-arrest phenotype of the double mutants, indicating that EIN3 plays an important role in regulating seedling growth and development [19]. In addition, studies have shown that EIL1 and EIN3 coordinate with each other in the ET regulation pathway [19].

EIN3/EIL is a key component of the ET signaling pathway and serves as a mediator for communication and interaction between ET and other hormone signals. The treatment of the *ein3 eil1* double mutant with jasmonate (JA) revealed varying degrees of inhibition in root hair and root development, with overexpression leading to enhanced root hair growth. This suggests that EIN3 and EIL1 act as positive regulators of the JA response [20,21]. During apical hook formation, gibberellins (GAs) work in conjunction with ET, with GA3 inducing HOOKLESS 1 (HLS1) and DELLA protein expression in an EIN3/EIL-dependent manner to jointly regulate hook curvature [22]. It can also interact with JA, salicylic acid (SA) and abscisic acid (ABA) to improve the plant response to biotic stress [23].

The study of EIN3/EIL gene functions is mainly concentrated on annual herbaceous plants, while the study of EIN3/EIL gene functions in woody plants is rarely reported. Haploid (H) or doubled haploid (DH) poplar has the advantages of a homozygous genome, clear genetic background and high genetic variation rate, and it is an ideal material for studying the gene functions of forest trees [24]. In this study, the EIN3/EIL transcription factor family was identified from the genome of DH poplar for the first time, and its sequence characteristics were analyzed. By exploring the expression characteristics of multiple EIN3/EIL genes in different DH plants and different tissues, the possible functions of the EIN3/EIL transcription factor family in the growth and development of DH were analyzed. Through the genetic transformation of *PsnEIL1* into *Populus simonii* × *P. nigra* (donor tree, DT), it was preliminarily confirmed that *PsnEIL1* can affect its growth and development and can cause the expression of downstream genes. This study provides an important theoretical basis to further explore the effect of the EIN3/EIL transcription factor family on the growth differences of different DH plants.

## 2. Results

### 2.1. Identification of PsnEIL Genes in DH Poplar

In order to identify the members of the EIN3/EIL transcription factor family in DH poplar, a combination of sequence similarity and conserved domain approaches was utilized, resulting in the identification of seven candidate *PsnEIL* genes. These genes were designated as *PsnEIN3*, *PsnEIL1*, *PsnEIL2*, *PsnEIL3a*, *PsnEIL3b*, *PsnEIL4* and *PsnEIL5* based on their homology to genes in *Arabidopsis thaliana* and *Populus trichocarpa* (Appendix A). The genomic locations of these seven *PsnEIL* genes were determined to be on six of the 19 chromosomes of DH poplar (Figure 1). Among them, only chromosome 1 contains two *PsnEIL* genes (*PsnEIL2* and *PsnEIL3b*), with only one gene on chromosome 3, chromosome 4, chromosome 8, chromosome 9 and chromosome 10, which were *PsnEIL3a*, *PsnEIN3*, *PsnEIL4*, *PsnEIL1* and *PsnEIL5*, respectively. Characteristic analysis was conducted on the identified genes, encompassing various parameters, such as the chromosome location, gene length, CDS length, number of amino acids, number of exons, molecular weight (Mw), isoelectric point (pI) and subcellular localization prediction data (Appendix A). The gene length varied from 1243 to 4327 bp, with the largest protein encoded by PsnEIL4, consisting of 662 amino acids, a CDS length of 1989 bp and a molecular weight of 74,934.21 kDa. In contrast, the smallest protein encoded by PsnEIL3b contained 207 amino acids, a CDS length of 624 bp and the smallest molecular weight of 23,486.95 kDa. The pI of these proteins varied from 5.12 (*PsnEIL2*) to 6.66 (*PsnEIL3b*). All PsnEIL proteins were predicted to localize to the nucleus (Appendix A).

### 2.2. Phylogenetic Analysis of EIN3/EIL Gene Family

In order to understand the phylogenetic relationships of the EIN3/EIL family, we constructed a maximum likelihood phylogenetic tree using IQ-TREE v1.6.12 [25] for a total of 48 EIN3/EIL protein sequences from DH poplar (seven genes), *Arabidopsis thaliana* (six genes), *Populus trichocarpa* (seven genes), *Salix purpurea* (six genes), *Vitis vinifera* (four genes), *Zea mays* (nine genes) and *Oryza sativa* (nine genes). The results indicated that the EIN3/EIL proteins were categorized into three distinct groups (A, B and C). Group A was found to lack any EIN3/EIL proteins from DH poplar, as well as from closely related species, such as *P. trichocarpa* and *S. purpurea*. Instead, Group A exclusively consisted of 10 EIN3/EIL proteins from *A. thaliana*, *V. vinifera*, *Z. mays* and *O. sativa*. This divergence may be attributed to the delayed speciation of Salicaceae species, as well as early gene replication events within the EIN3/EIL gene family, leading to functional divergence and the formation of Group A. There were 14 and 24 EIN3/EIL proteins in group B and group C, respectively. Among them, PsnEIL2, PsnEIL3a and PsnEIL3b belonged to group B, and PsnEIL1, PsnEIN3, PsnEIL4 and PsnEIL5 belonged to group C. The EIN3/EIL proteins of all species were distributed in both group B and group C (Figure 2).

### 2.3. Conserved Motif and Gene Structure Analysis of PsnEIL Genes

In order to understand the structural differences of the *PsnEIL* genes in DH poplar, we analyzed the conserved motifs and gene structure of the *PsnEIL* genes and constructed a phylogenetic tree of seven PsnEIL proteins using the maximum-likelihood method to further explore the differences in gene structure and evolutionary relationships. The phylogenetic analysis showed that the PsnEIL proteins were still divided into two groups, and the results were consistent with the above phylogenetic analysis results of the multiple-species EIN3/EIL family (Figure 3a). The results of conserved motif analysis indicated that there were similar distributions of motifs on the same evolutionary branch; for example, in evolutionary branch II, the PsnEIL1, PsnEIN3, PsnEIL4 and PsnEIL5 protein sequences all distributed 10 motifs, and the order of each motif in the respective protein sequence was consistent. However, in evolutionary branch I, the composition of motifs on PsnEIL2, PsnEIL3a and PsnEIL3b differed, but all contained Motif 4, Motif 1 and Motif 5 (Figure 3b). In the analysis of gene structure, it was observed that the *PsnEIL2* gene contains five exons, while the remaining genes typically have one or two exons. Specifically, *PsnEIL3a* and *PsnEIL3b* possess only one intron, while *PsnEIL1*, *PsnEIN3*, *PsnEIL4* and *PsnEIL5* are single-exon genes without introns. Furthermore, only *PsnEIL3a* and *PsnEIL4* exhibit 5′-UTR and 3′-UTR regions, whereas the other genes lack these regions due to incomplete genome annotation. Additionally, each gene contains a complete EIN3 domain and is situated closer to the 5’ end of the sequence (Figure 3c).

### 2.4. Analysis of Cis-Acting Elements of PsnEIL Promoters

The promoter sequence located upstream of the gene-coding sequence has many cis-acting elements that are used to regulate gene-specific expression. To enhance comprehension of the potential function of *PsnEIL* genes, an analysis was conducted on the 2000 bp promoter sequences located upstream of *PsnEIL* genes, revealing the presence of 19 distinct types of cis-acting elements (Figure 4 and Appendix A). Notably, light-responsive regulatory elements were identified on the promoters of all genes, along with phytohormone-responsive response elements associated with auxin, abscisic acid, gibberellin, salicylic acid and methyl jasmonate (MeJA) responses. Among them, ABRE elements related to the abscisic acid response were found in the promoters of *PsnEIL1*, *PsnEIL2*, *PsnEIN3* and *PsnEIL4*, elements related to the auxin response were found in the promoters of *PsnEIL3a*, *PsnEIL4* and *PsnEIL5* and elements related to the gibberellin response were found in the promoters of *PsnEIL1*, *PsnEIL3a*, *PsnEIL3b*, *PsnEIL4* and *PsnEIL5*. In the promoter region of *PsnEIN3*, there are several other phytohormone-responsive elements, in addition to auxin- and gibberellin-response elements. All *PsnEIL* gene promoters contain cis-acting regulatory elements related to plant growth, including circadian control, zein metabolism, meristem expression, endosperm expression and palisade mesophyll cell differentiation. A circadian control element was found in the promoter region of *PsnEIL5*, and CAT-box elements related to meristem expression were found in the promoter regions of *PsnEIL2* and *PsnEIL3a*. Regarding the stress response, ARE elements related to anaerobic induction were present in the promoter regions of other genes, except *PsnEIL2*, TC-rich repeats elements related to defense and the stress response were present in the promoter regions of *PsnEIL1*, *PsnEIL2*, *PsnEIL3b*, *PsnEIL4* and *PsnEIL5*, LTR elements related to the low-temperature response were present in the promoter regions of *PsnEIL1* and *PsnEIL3a* and MBS elements related to drought induction were present in the promoter regions of *PsnEIL1*, *PsnEIL3a* and *PsnEIL3b*. The above results indicate that *PsnEIL* genes have potential roles in photosynthesis, growth and development, plant hormone responses and stress responses.

### 2.5. Chromosome Distribution and Synteny Analysis of PsnEIL Genes

Gene duplication events are a common occurrence in plant gene family formation, playing a crucial role in the adaptive evolution of species. In order to investigate the duplication events of all *PsnEIL* genes in DH poplar, a synteny analysis was conducted using MCscanX and visualized with Advanced Circos in TBtools v1.132 [26,27]. The analysis revealed a single tandem duplication event involving the gene pair *PsnEIL2* and *PsnEIL3b* on chromosome 01 (Figure 5 and Appendix A). Moreover, our study identified two gene pairs exhibiting segmental duplication events. Gene selection pressure analysis was conducted using TBtools v1.132 to determine the non-synonymous substitution (Ka) and synonymous substitution (Ks) values of the EIN3/EIL gene family segmental duplication and tandem duplication gene pairs in DH poplar. The analysis revealed that the Ka/Ks ratios of *PsnEIL2* and *PsnEIL3b*, *PsnEIL4* and *PsnEIL5* and *PsnEIL2* and *PsnEIL3a* were 0.45, 0.29 and 0.36, respectively. The fact that all values were less than 1 suggests that the EIN3/EIL gene family may have undergone significant purification selection pressure throughout its evolutionary history (Appendix A).

### 2.6. Syntenic Relationships of the EIN3/EIL Gene Family between DH Poplar and Other Species

In order to investigate the synteny relationships between *PsnEIL* genes and related genes from the other six representative species, including four eudicots (*Arabidopsis thaliana*, *Populus trichocarpa*, *Salix purpurea* and *Vitis vinifera*) and two monocots (*Oryza sativa* and *Zea mays*), a comparative synteny analysis was conducted. The numbers of orthologous gene pairs between DH poplar and various plant species, including *A. thaliana*, *P. trichocarpa*, *S. purpurea*, *V. vinifera*, *O. sativa* and *Z. mays*, were determined to be 4, 11, 11, 4, 4 and 4, respectively (Figure 6). These findings suggest that DH poplar exhibits a higher degree of homology with *S. purpurea* and *P. trichocarpa* compared to other species, indicating a relatively conserved evolutionary relationship within the same genus.

### 2.7. Expression Pattern Analysis of PsnEIL Genes in Different DH Plants

Doubled haploid (DH) poplars are generated through anther culture, in vitro, of diploid wild-type *Populus simonii* × *P. nigra* (donor tree, DT). Despite their origin from DT, DH plants exhibit notable phenotypic distinctions from the DT, as well as significant variations among themselves (Figure 7a). These distinctions encompass traits, such as the plant height, stem development and survival rate. The DH poplar population displays a pronounced level of genetic diversity, as evidenced by the unique characteristics of individual DH plants, as previously highlighted in our research [24,28]. To delve deeper into the impact of *PsnEIL* genes on various DH plants, we conducted an analysis of their expression in the apical buds of DT and four distinct DH plants (DH1207, DH1716, DH1717 and DH1588), utilizing the RNA-seq data from a previous study [28] (Figure 7b and Appendix A). The findings indicated that the expression levels of *PsnEIL1*, *PsnEIL2*, *PsnEIL3b*, *PsnEIL4* and *PsnEIL5* genes were markedly elevated in the apical buds of DH1716 and DH1717, with only the *PsnEIN3* gene showing higher expression in DH1717. Conversely, the expression level of the *PsnEIL3a* gene was higher in other DH plants and DT, with the exception of lower expression in DH1588 and DH1716. Furthermore, the expression levels of the remaining genes in the EIN3/EIL transcription factor family, excluding the *PsnEIL3a* gene, were notably lower in DT. As evidenced by the growth variations observed among various DH plants (Figure 7a) [28], the findings above suggest that the *PsnEIL* genes may play a role in influencing the growth and development of diverse DH plant species.

### 2.8. Tissue-Specific Expression Analysis of PsnEIL Genes in Different DH Plants

To investigate the variations in the expression of *PsnEIL* genes in DH plants and DT tissues, we conducted an analysis of the expression levels of *PsnEIL1*, *PsnEIN3*, *PsnEIL4* and *PsnEIL5* genes in apical buds, young leaves, functional leaves, xylem, cambium and roots of DT, DH1588 and DH171717 using RT-qPCR (Figure 8). The findings indicated the significant upregulation of *PsnEIL1* expression in the xylem and root tissues of DT plants compared to in other tissues, with the exception of the cambium. Similarly, in DH1588 and DH1717 plants, *PsnEIL1* expression in the xylem and root tissues was elevated relative to that in the apical bud and leaf tissues, resembling the pattern observed in DT plants. Notably, the expression level of *PsnEIL1* in the xylem tissue of DT plants was approximately three times higher than that in DH1588 and DH1717 plants. The expression levels of *PsnEIN3* varied across different tissues in DT, DH1588 and DH1717. Specifically, *PsnEIN3* exhibited significantly higher expression levels in the functional leaves, xylem, cambium and roots of DT compared to that in the apical buds and young leaves. In DH1588 and DH1717, the expression levels of *PsnEIN3* in xylem and cambium were notably elevated in comparison to those in young leaves and functional leaves. Furthermore, in DH1717, the expression level of *PsnEIN3* in apical buds surpassed that in DT and DH1588, aligning with the observations presented in Figure 7b. The expression levels of *PsnEIL4* were found to be low in all tissues of DT and DH1717. However, in the young leaves of DH1588, the expression of *PsnEIL4* was significantly elevated compared to that in other tissues, with levels approximately 30 times higher than those observed in the young leaves of DT and DH1717. Furthermore, in DH1717, the expression of *PsnEIL4* in apical buds was notably higher than in DT and DH1588, consistent with the findings presented in Figure 7b. The expression level of *PsnEIL5* in various tissues of DT, DH1588 and DH1717 varied significantly. Specifically, *PsnEIL5* exhibited significantly higher expression levels in the young leaves, functional leaves and roots of DT compared to in the apical buds, xylem and cambium. In DH1588, the expression level of *PsnEIL5* was notably higher in the xylem and roots compared to in other tissues. Additionally, the expression level of *PsnEIL5* in apical buds of DH1717 was the highest among the three genotypes, surpassing the expression levels in apical buds of DT and DH1588. The findings presented above indicate notable variations in the expression levels of *PsnEIL* genes across various tissues of distinct DH plants, implying potential functional disparities within the EIN3/EIL gene family across different DH plants and underscoring their crucial involvement in the growth and development of such plants.

### 2.9. Overexpressing PsnEIL1 Gene in DT

In order to explore the effect of the *PsnEIL1* gene on poplar growth and development, especially secondary growth, we overexpressed *PsnEIL1* in DT under the control of a CaMV 35S promoter. Six independent transgenic lines (*OE-PsnEIL1-L2*, *-L43*, *-L56*, *-L65*, *-L36* and *-L37*) were generated, and the expression levels of *PsnEIL1* were significantly increased in all transgenic lines (Figure 9a). Among them, the expression level in the *OE-PsnEIL1-L43* transgenic line was the highest, which was about 60 times that of the wild-type, and it showed stunted growth relative to the wild-type (Figure 9b). From the above transgenic lines, we selected *OE-PsnEIL1-L43*, *OE-PsnEIL1-L36* and *OE-PsnEIL1-L56* transgenic lines and carried out plant height statistical analysis. The results showed that they were significantly lower than the wild-type, and because the *PsnEIL1* transcription level in the *OE-PsnEIL1-L43* plant was the highest, its plant height was the lowest (Figure 9c). Then, the downstream target genes *PsnERF33*, *PsnERF34* and *PsnERF102* of *PsnEIL1* in transgenic *OE-PsnEIL1-L43* plants were analyzed via RT-qPCR. The results showed that the expression levels of these three genes in *OE-PsnEIL1-L43* plants were significantly higher than those in the wild-type, which was consistent with previous studies [29]. In summary, the *PsnEIL1* gene may be involved in regulating the secondary growth of poplar and affecting the growth and development of different DH plants.

## 3. Discussion

Transcription factors (TFs) are a class of protein molecules that can specifically bind to cis-acting elements in gene promoter regions [30]. They regulate the transcription and expression of target genes by interacting with cis-acting elements of downstream target genes, which plays an important role in plant growth, development and stress responses [30,31,32,33]. The EIN3/EIL gene family is an important gene family. The EIN3/EIL genes affect plant growth and development by participating in the ET signal transduction process [34]. Moreover, EIN3 and EIL1 not only participate in the ET signal transduction pathway, but also serve as the integration center of ET and other signals, thereby extensively regulating plant growth, development and resistance to stress [20,21,22,23,35]. This study used doubled haploid (DH) poplar genome data and screened DH poplar EIN3/EIL family proteins based on the conservation of EIN3/EIL protein sequences. Finally, seven EIN3/EIL transcription factor genes were identified in DH poplar and named based on their homology with *A. thaliana and P. trichocarpa*.

Intron loss and gain are major phenomena in the evolution of gene structure [36]. Sequence changes in introns have certain effects on the stability, localization and translation of transcripts, which may be manifested in different developmental stages or environmental stresses [37,38]. Therefore, this study predicted the gene structure of the EIN3/EIL gene family and found that four of the seven genes (*PsnEIL1*, *PsnEIN3*, *PsnEIL4* and *PsnEIL5*) did not contain introns. Three genes contain introns, and two genes (*PsnEIL3b* and *PsnEIL3a*) have one intron region. It is worth noting that the *PsnEIL2* gene contains four intron, suggesting that this may be the cause of functional differences between genes. Collinearity analysis of DH and EIN3/EIL homologous genes of the other six plants was performed. These EIN3/EIL genes were divided into three branches (Figure 2). It is worth noting that the EIN3/EIL genes of DH are only in the B and C clades, presumably because they are assigned to these two clades because of similarities in gene structure and function. Subsequently, this study analyzed the expression pattern of EIN3/EIL genes in DH. EIN3/EIL genes were expressed in most tissue samples. Only *PsnEIL4* was significantly highly expressed in the leaves of DH1588 plants. It is speculated that the spatiotemporal expression differences are related to the functional differences in the genes.

In order to further explore the function of the *PsnEIL1* gene, the CDS sequence of *PsnEIL1* was cloned from DH poplar, and a plant expression vector (*35S:PsnEIL1*) was constructed. *OE*-*PsnEIL1* transgenic plants were obtained. Through the analysis of transgenic plants, it was found that the plants became shorter. In this study, it was speculated that this phenotype may be directly caused by the overexpression of *PsnEIL1*, which is similar to the results of previous studies [19]. However, its specific functional mechanism still needs further experimental verification. ET has dual effects on plant development, both promoting and inhibiting it [39]. Impairment of ET response pathways reduces tolerance or resistance to environmental stress, but the overactivation of ET signaling can lead to growth inhibition and even plant death. Therefore, it is necessary to explore the function of the EIN3/EIL gene family. This study provides candidate genes for the poplar EIN3/EIL1 gene family in regulating poplar growth and provides a reference at the molecular level.

## 4. Materials and Methods

### 4.1. Plant Materials and Growth Conditions

Diploid wild-type *Populus simonii* × *P. nigra* (donor tree, DT) and doubled haploid plants DH1588, DH1717, DH1716 and DH1207 were used for all experiments. Among them, these different DH lines were obtained via anther culture, in vitro, of the DT [24]. All plants were grown in a walk-in growth chamber. DT was used for transgenic experiments. Samples were collected from various tissues of transgenic plant leaves and 3-month-old DH plants, including apical buds, young leaves, functional leaves, xylem, cambium and roots. The samples were promptly frozen in liquid nitrogen and stored at −80 °C for subsequent RT-qPCR analysis.

### 4.2. Identification of the Psn/EIL Genes in DH Poplar

We identified the EIN3/EIL gene family in the DH poplar genome using sequence similarity and conserved domain approaches. The DH poplar genome from *Populus simonii* × *P. nigra* has been sequenced by our research group. The known EIN3/EIL amino acid sequences in *Arabidopsis*, obtained from The Arabidopsis Information Resource (TAIR) (https://www.arabidopsis.org/, accessed on 21 April 2023) [40], were used to search the DH poplar genome using BLASTP with an e-value cutoff of 1 × 10^−5^. The EIN3 domain HMM profile (PF04873) was obtained from the InterPro database (https://www.ebi.ac.uk/interpro/, accessed on 21 April 2023) [41] and utilized to search the genome protein databases with an e-value cutoff of 1 × 10^−5^ using HMMER v3.3.2 software [42]. To confirm the presence of the EIN3 conserved domain in the candidate EIN3/EIL genes, the Batch CD-Search Tool (https://www.ncbi.nlm.nih.gov/Structure/bwrpsb/bwrpsb.cgi, accessed on 21 April 2023) and SMART (http://smart.embl.de/, accessed on 21 April 2023) were utilized for verification. Molecular characteristics and subcellular localization were analyzed using TBtools v1.132 [26] and DeepLoc-2.0 (https://services.healthtech.dtu.dk/services/DeepLoc-2.0/, accessed on 30 April 2023) [43], respectively.

### 4.3. Chromosomal Location of PsnEIL Genes

The *PsnEIL* genes were identified on the chromosome of DH poplar through an analysis of the DH poplar genome annotation file in either gff3 or gtf format, utilizing TBtools v1.132 [26].

### 4.4. Phylogenetic Analysis

In order to explore the phylogenetic relationship of the EIN3/EIL gene family, we downloaded the genome sequences and annotation files of *Arabidopsis* (Araport11) [44], *Populus trichocarpa* (v4.1) [45], *Salix purpurea* (v5.1) [46], *Vitis vinifera* (v2.1) [47], *Zea mays* (Zm-B73) [48] and *Oryza sativa* (v7.0) [49] from Phytozome v13 (https://phytozome-next.jgi.doe.gov/, accessed on 10 May 2023). Using the methods mentioned above, we identified the EIN3/EIL gene family members of these species and constructed a phylogenetic tree with DH poplar.

We used the Muscle program in MEGA11 [50] to perform multiple sequence alignments of the EIN3/EIL genes of DH poplar and six other species and then used IQ-TREE v1.6.12 [25,51] to create a maximum likelihood phylogenetic tree with 5000 ultrafast replications and used iTOL v6.7.1 (https://itol.embl.de/, accessed on 10 May 2023) [52] to beautify the phylogenetic tree. The phylogenetic tree of seven PsnEIL proteins in DH poplar was constructed according to the above method.

### 4.5. Conserved Motifs and Gene Structure Analysis

The conserved motifs of *PsnEIL* genes were identified through the utilization of MEME v5.5.1 (https://meme-suite.org/meme/tools/meme, accessed on 20 May 2023), with the number of conserved motifs set to 10 and default parameters utilized [53]. The gene structure of *PsnEIL* genes was analyzed based on the DH poplar genome annotation file using TBtools v1.132, and the visualization of the phylogenetic tree, conserved motifs, conserved domains and gene structure merging were conducted.

### 4.6. Cis-Elements Analysis

The 2000 bp sequence upstream of the CDS of *PsnEIL* genes was obtained using TBtools v1.132 [26], as the promoter sequence, and the cis-acting elements on the *PsnEIL* promoters were analyzed using the PlantCARE website (https://bioinformatics.psb.ugent.be/webtools/plantcare/html/, accessed on 10 May 2023) [54].

### 4.7. Expression Pattern Analysis

To investigate the expression patterns of PsnEIL genes in various DH plants, we conducted an analysis based on RNA-seq data from prior research [28]. Specifically, we examined the expression of PsnEIL genes in the apical buds of diploid wild-type *Populus simonii* × *P. nigra* (donor tree, DT) and four distinct DH plants (DH1207, DH1716, DH1717 and DH1588), presenting the results visually in a heat map. The raw data can be found in Appendix A. Concurrently, we employed RT-qPCR to examine the tissue-specific expression patterns of *PsnEIL* genes in apical buds, roots, xylem, cambium and leaves across DH plants DH1588, DH1717 and diploid wild-type hybrids of *Populus simonii* × *P. nigra* (donor tree, DT).

### 4.8. Generation of Populus simonii × P. nigra Transgenic Plants

The construction of a plant overexpression vector referred to the previous method [55]. In short, the full-length coding sequence of *PsnEIL1* was confirmed through sequencing and inserted into the pBI121 binary vector driven by the cauliflower mosaic virus 35S (CaMV 35S) promoter to generate an overexpression construct [56]. The recombinant plasmid was introduced into *Agrobacterium tumefaciens* strain GV3101 for *Populus simonii* × *P. nigra* transformation using the leaf disc method. The transformation steps were as follows: *Agrobacterium tumefaciens* (GV3101) carrying the pBI121-*PsnEIL1* vector plasmid was cultured at 28 °C and 200 rpm until the OD_600_ (optical density at 600 nm) value was 0.6–0.8. The cells were collected via centrifugation and resuspended with the same volume of sterile water. At the same time, 70 μM acetosyringone (AS) and 0.002 mg/L thidiazuron (TDZ) were added to prepare the transformation solution. The pruned sterile leaves (1 cm × 1 cm) of *P. simonii* × *P. nigra* were placed in the transformation solution and co-cultured for 3–5 min. The leaves were taken out and transferred to sterile water containing 200 mg/L timentin for sterilization. After that, the leaves were tiled to the resistance screening medium (MS + 0.5 mg/L 6-BA + 0.1 mg/L NAA + 30 g/L sucrose + 7.5 g/L agar + 20 mg/L Kan). After one month, the resistant buds were grown and transferred to the stemming medium (MH + 0.2 mg/L 6-BA + 0.05 mg/L NAA + 30 g/L sucrose + 7.5 g/L agar + 25 mg/L Kan). After two weeks, the buds were transferred to rooting medium (1/2MS + 0.4 mg/L IBA + 0.02 mg/L NAA + 20 g/L sucrose + 7.5 g/L agar + 20 mg/L Kan) and cultured for one month to obtain rooting-resistant plants. The resistant plants were detected via a PCR reaction at the DNA level with the universal upstream and downstream primers CaMV 35S_F and M13F, so as to confirm the transgenic lines. The primers used for vector construction are given in Appendix A.

### 4.9. RNA Extraction and RT-qPCR

Total RNA was isolated from the apical buds, young leaves, functional leaves, xylem, cambium and roots of DH1588, DH1717, diploid wild-type *Populus simonii* × *P. nigra* (donor tree, DT) and the leaves of transgenic plants using a Universal Plant Total RNA Fast Extraction Kit (BioTeke, Beijing, China). The RNA concentration was quantified utilizing a NanoDrop 2000 spectrophotometer (Thermo Fisher Scientific, Waltham, MA, USA). Subsequently, 1 μg of total RNA from each sample was used for cDNA synthesis with the PrimeScript RT reagent Kit with gDNA Eraser (TaKaRa, Dalian, China) following the manufacturer’s instructions. RT-qPCR experiments were conducted using a 7500 Fast Real-Time PCR System (Applied Biosystems, Waltham, MA, USA) with TB Green Premix Ex Taq II (Tli RNaseH Plus) (TaKaRa). The reference gene *PsnACTIN* was utilized to assess the relative expression of the target gene, with three biological replicates and three technical replicates. Appendix A contains primers used in RT-qPCR.

### 4.10. Statistical Analysis

Student’s *t* tests were performed using the ‘Significance marker difference analysis (Multiple groups)’ tool in the free online data analysis platform OmicShare.

## 5. Conclusions

In this study, we identified a total of seven EIN3/EIL genes from DH poplar. We analyzed the chromosomal location, conserved domains, physicochemical properties, gene structure, phylogeny and microsynteny of these genes. At the same time, we used transcriptome data and RT-qPCR to verify the expression pattern of the EIN3/EIL gene in the terminal buds of DT and DH poplars, indicating that the EIN3/EIL gene is highly expressed in DH. Following this, we conducted an analysis of the expression patterns of EIN3/EIL genes in various tissues of different DH poplar, including apical buds, roots, xylem, cambium and leaves. Additionally, we performed the overexpression of *PsnEIL1* in DT, resulting in transgenic plants displaying a dwarf growth phenotype. These findings suggest that the *PsnEIL1* gene may be implicated in the regulation of secondary growth in poplar and impact the growth and development of DH poplar. In conclusion, the EIN3/EIL gene family may play a significant role in the growth and development of DH poplar.

## Figures and Tables

**Figure 1 ijms-25-04116-f001:**
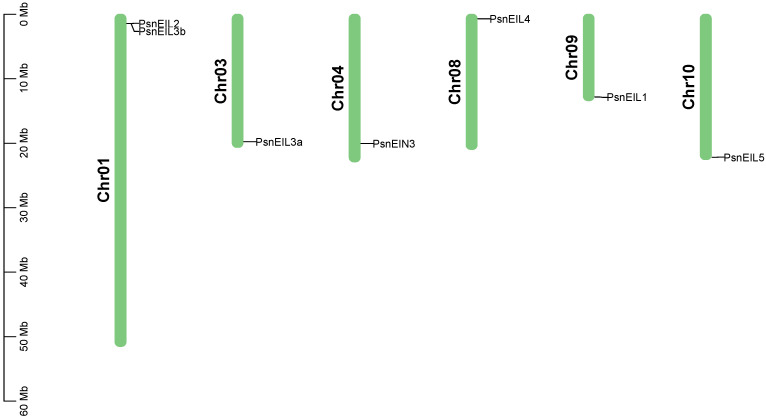
Distribution of *PsnEIL* genes on chromosomes of DH poplar.

**Figure 2 ijms-25-04116-f002:**
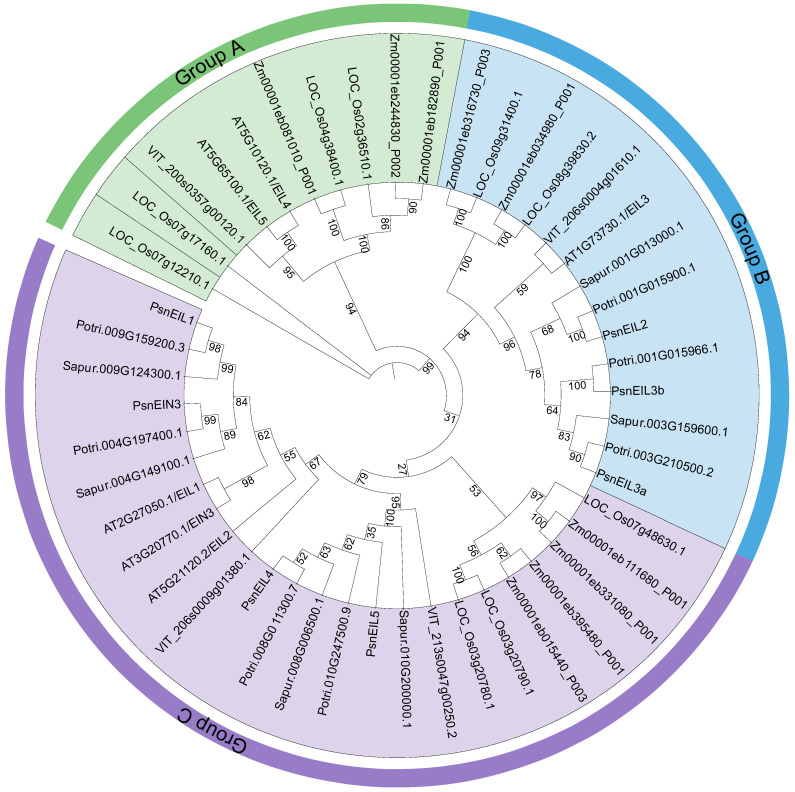
Maximum-likelihood phylogenetic tree of EIN3/EIL family proteins in DH poplar, *Arabidopsis thaliana*, *Populus trichocarpa*, *Salix purpurea*, *Vitis vinifera*, *Zea mays* and *Oryza sativa*. Branches and labels of different colors represent different groups, and the numbers at the nodes represent bootstrap values.

**Figure 3 ijms-25-04116-f003:**
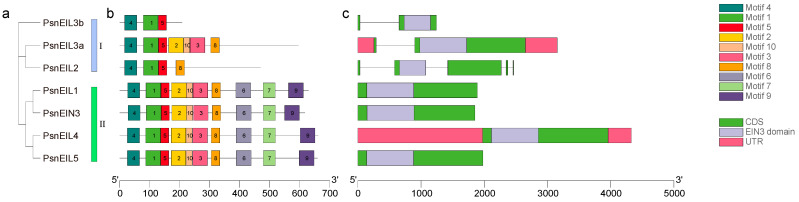
An analysis of the phylogenetic relationship, conserved motifs and gene structure of *PsnEIL* genes was conducted. (**a**) The phylogenetic tree of all PsnEIL proteins was constructed using the maximum-likelihood method. (**b**) The distribution of conserved motifs in PsnEIL proteins was examined, with motifs 1–10 represented as rectangular boxes of various colors. (**c**) Gene structures of *PsnEIL* genes were analyzed, with pink boxes indicating 5′ UTR and 3′ UTR regions, green boxes representing exons, gray lines representing introns and a purple box highlighting the EIN3 domain within the corresponding gene sequence.

**Figure 4 ijms-25-04116-f004:**
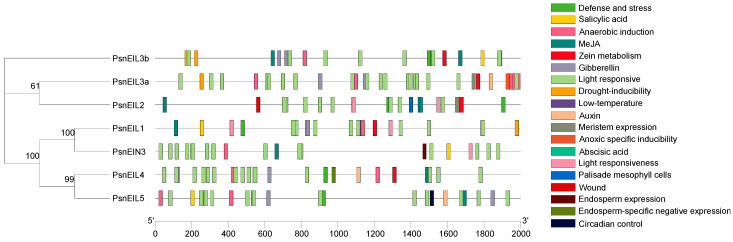
Analysis of cis-acting elements in the promoter region of the *PsnEIL* genes.

**Figure 5 ijms-25-04116-f005:**
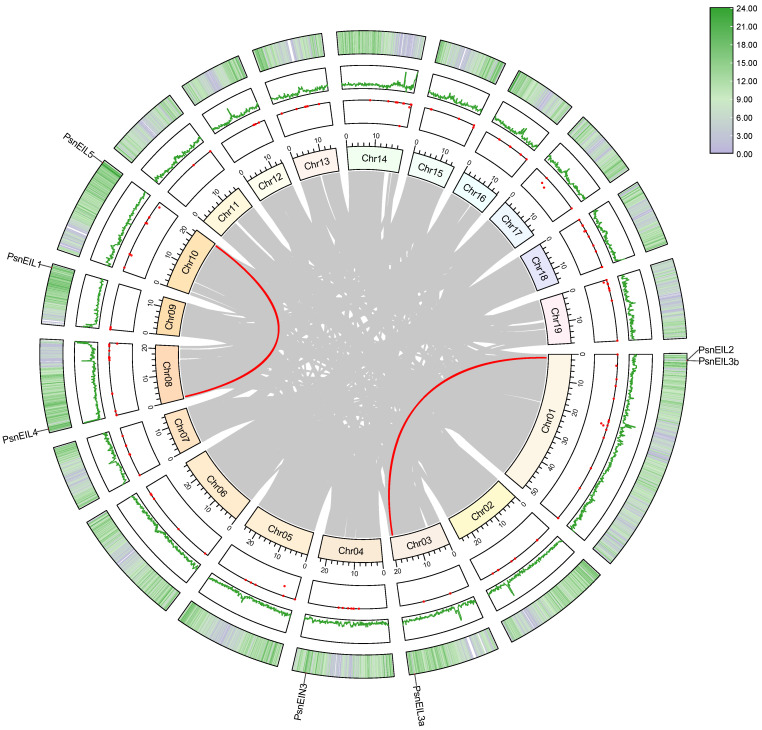
Collinearity analysis of the EIN3/EIL gene family in DH poplar. The visualization includes a gradient color rectangle representing chromosomes 01–19, a red dot indicating gap distribution on the genome, a green line representing the GC ratio on the genome, and a heat map on the outermost circle illustrating gene density. Synteny blocks in the DH poplar genome are denoted by gray lines in the center, while segmental duplication gene pairs are delineated by red lines between chromosomes.

**Figure 6 ijms-25-04116-f006:**
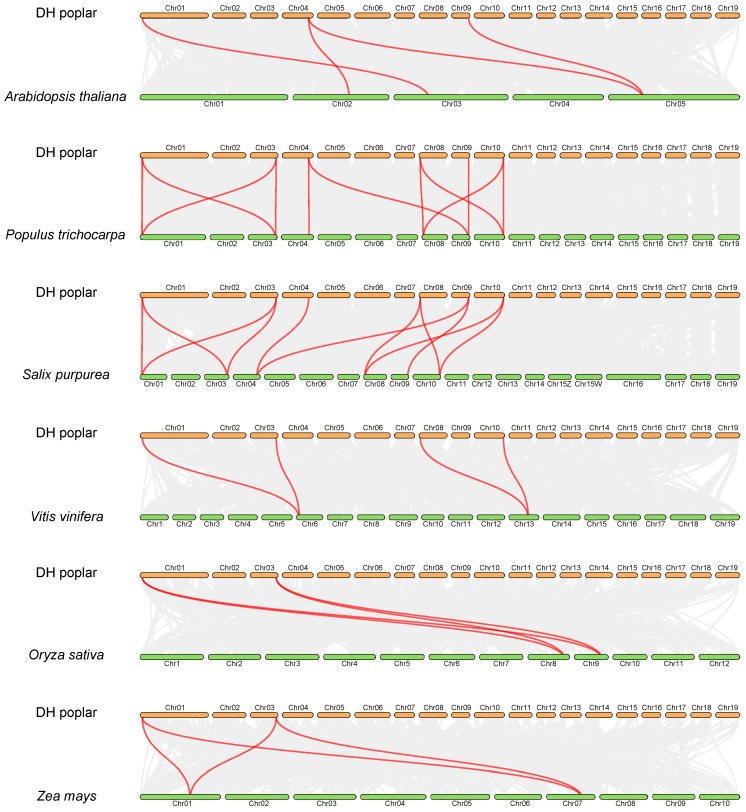
The synteny analysis of *PsnEIL* genes was conducted between DH poplar and six other plant species, with gray lines representing gene blocks in DH poplar that are orthologous to the other genomes and red lines delineating the syntenic EIN3/EIL gene pairs.

**Figure 7 ijms-25-04116-f007:**
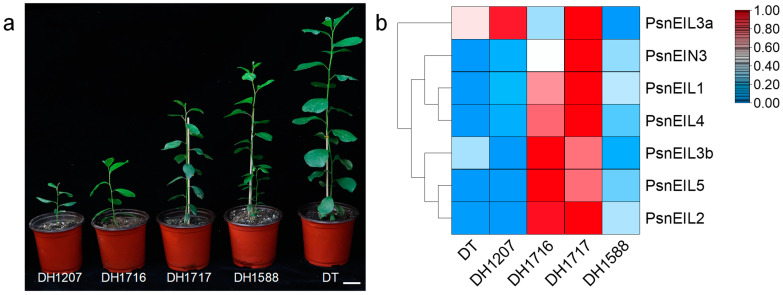
Expression pattern analysis of *PsnEIL* genes in different DH plants. (**a**) Growth phenotypes of different DH plants. Bars = 5 cm. DH1207: doubled haploid line 1207, DH1716: doubled haploid line 1716, DH1717: doubled haploid line 1717, DH1588: doubled haploid line 1588, DT: donor tree (diploid wild-type *Populus simonii* × *P. nigra*). (**b**) The transcript abundance of *PsnEIL* genes in the apical buds of four DH plants was analyzed via RNA-seq.

**Figure 8 ijms-25-04116-f008:**
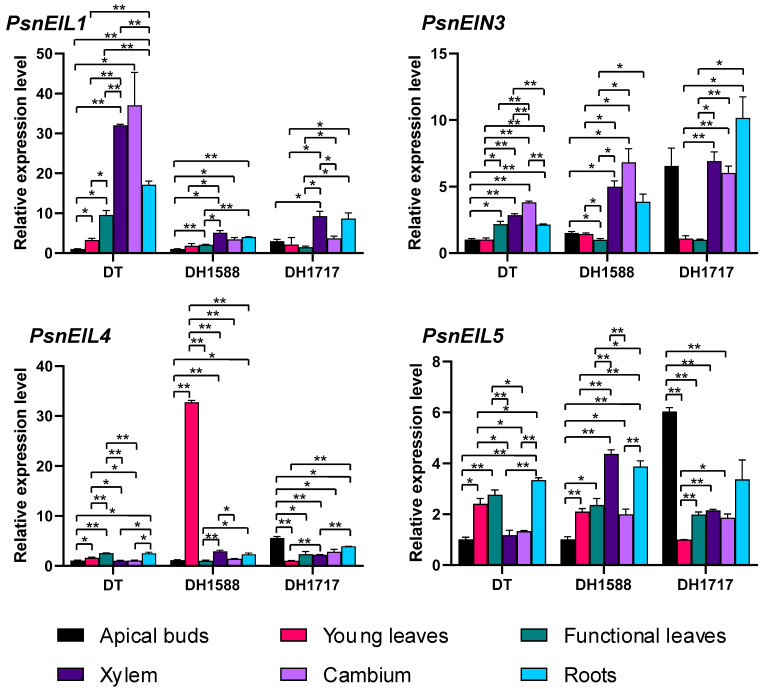
Expression analysis of *PsnEIL1*, *PsnEIN3*, *PsnEIL4* and *PsnEIL5* genes was conducted using various tissues, including apical buds, young leaves, functional leaves, xylem, cambium and roots of DT, DH1588 and DH1717. DT: donor tree (diploid wild-type *Populus simonii* × *P. nigra*), DH1588: doubled haploid line 1588, DH1717: doubled haploid line 1717. Error bars represent the SE values of three biological replicates. Asterisks indicate significant differences in the expression levels of *PsnEIL* genes among different tissues based on a Student’s *t* test (*, *p* < 0.05, **, *p* < 0.01).

**Figure 9 ijms-25-04116-f009:**
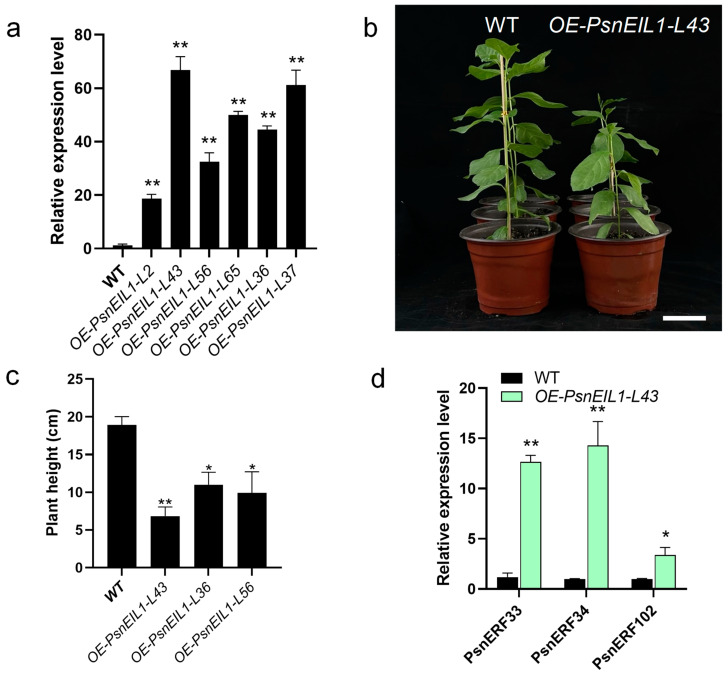
Overexpression of the *PsnEIL1* gene in DT. (**a**) Expression levels of the *PsnEIL1* gene in six *PsnEIL1* transgenic lines (*OE-PsnEIL1-L2*, *-L43*, *-L56*, *-L65*, *-L36* and *-L37*). *PsnACTIN* was used as an internal reference gene. Error bars represent the SE values of three biological replicates. Asterisks indicate significant differences between each line of the transgenics and wild-type (WT) plants based on a Student’s *t* test (**, *p* < 0.01). (**b**) Growth phenotypes of 1-month-old wild-type and *OE-PsnEIL1-L43* transgenic plants. Bars = 5 cm. (**c**) Statistical analysis of height of wild-type (WT) and *OE-PsnEIL1* transgenic plants (*OE-PsnEIL1-L43*, *-L36* and *-L56*). Error bars represent the SE values of three biological replicates. Asterisks indicate significant differences between different lines of transgenic plants and wild-type (WT) plants based on a Student’s *t* test (*, *p* < 0.05, **, *p* < 0.01). (**d**) Expression levels of the *PsnERF33*, *PsnERF34* and *PsnERF102* genes in wild-type (WT) and *OE-PsnEIL1-L43* transgenic plants. *PsnACTIN* was used as an internal reference gene. Error bars represent the SE values of three biological replicates. Asterisks indicate significant differences between transgenics and wild-type (WT) plants based on a Student’s *t* test (*, *p* < 0.05, **, *p* < 0.01).

## Data Availability

Data are contained within the article and Appendix A.

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
