# Peer review of "Genome-Wide Identification and Analysis of the EIN3/EIL Transcription Factor Gene Family in Doubled Haploid (DH) Poplar"

_ijms, 2024, doi:10.3390/ijms25074116_

Round 1

Reviewer 1 Report

Comments and Suggestions for Authors

In this manuscript Liu and collaborators perform an in-silico genome-wide characterization of EIN3/EIL TF family in a double haploid poplar accession and used RNAseq data to investigate the expression profiles of the family members they identified. Last, to assess the activity of EIL1, they perform genetic transformation. With this regard, you should discuss the reason why you decided to assess EIL1: I can imagine why, but you should add it anyway.

The manuscript is interesting and gives an overview and an albeit preliminary functional characterization. Interesting are the diverse expression levels of such TFs in the different lines: because of such difference, I would expect first a description of such lines, and then a discussion about the results you obtained. More in general, the discussion should be expanded by considering all the results (if deemed worth) you obtained.

- what does "Psn" stand for? Please clarify

- line 67: "EIN3 and EIL1 have most closest homolog related": you mean "are the closest"?

- section 2.3: do the authors check whether conserved motifs actually have a function?

- line 148: Figure 3a

- section 2.8: in the text you state that the differences were "significant", meaning that you performed statistics on that. This part is missing in the methods as is missing, in the related Figure (Fig. 8) a tag to identify such genes. Please add related P-value as well.

- section 2.9 and related discussion: EIL1 expression is higher than in wt (is it DT?), apparently similar to what you evidenced for DH1716 and DH1717 (Fig. 7): are they phenotypes comparable?

- section 4.8 should be deeper described, as very few info are given: did you use a negative control? How did you assess the success of the transformation? Did you regenerate plants or what else? It would also be useful to add the promoter present to drive the expression of the transgene

- Figure 4: please add in the methods how you obtained the dendrogram

- Figure 8: the figure is not very clear, as it is not possible to appreciate (eventual) differences. Error bars are missing. Moreover the caption has to be modified: indicate which panel refers to which gene, and remove the blue/red part.

- Figure 9c: if you have, you should add the plant height of all the transgenic lines you introduced in panel a

Comments on the Quality of English Language

please carefully revise English language as there are parts throughout the text that need to be edited (for instance: "It’s have various functions")

Author Response

Dear Reviewer,

We appreciate your thorough review and insightful suggestions for our paper. We have addressed and elaborated on your inquiries and recommendations in the revised draft and response document.

Thank you once again for your meticulous review and valuable feedback.

Sincerely,

Caixia Liu

Reviewer 2 Report

Comments and Suggestions for Authors

Review of ijms-2925532

Genome-Wide Identification and Analysis of the EIN3/EIL Transcription Factor Gene Family in Doubled Haploid Poplar

Caixia Li, , Erqin Fan, Yuhang Liu, Meng Wang, Qiuyu Wang, Sui Wang, Su Chen, Chuanping Yang, Xiangling You and Guanzheng Qu

The authors used bioinformatics tools to search the polar genome for members of the EIN3/EIL gene family and found 7.  They then performed routine bioinformatics analyses and found that all were predicted to localize in the nucleus and that the poplar genes could be divided into 2 groups with similar structures and conserved motifs.  They next studied expression of these genes in various plant tissues in order to study the issue of gene redundancy, and found using RT-qPCR that PsnEIL4 was only expressed in leaves while PsnEIL3b was highly expressed in the apical buds. They concluded by overexpressing PsnEIL1 in transgenic plants, and found that the resulting plants were dwarf.

The work seems to have been performed using suitable techniques with adequate replication and suitable statistical analysis except for figure 8.  I therefore feel that it will be suitable for publication after correcting the issues noted below.

My biggest concern is that I would like the authors to specifically state how their data can improve poplar breeding. What genes should breeders focus on to improve poplar?

I am also troubled by their statement on lines 276-277 that EIN3/EIL genes play an important role in regulating the growth and development of DH plants. They have shown correlation, not causation. Please rephrase to reflect this.

The caption for figure 8 (lines 279-281) does not match what is shown in the figure. Please delete “Red represents high expression level and blue represents low expression level.”   Please state the number of biological and technical replicates, whether any of these differences were statistically significant and at what level. Also please state the reference gene.

Section 4.8: please provide a more detailed description of how the transgenic plants were created and selected for.

Section 4.9: please state the reference gene used for RT-qPCR

Specific comments

The English has lots of mistakes and is very hard to understand in many places. Specific examples are listed below, but there are many more. I recommend editing by a native English speaker before resubmitting.

Lines 18-19: as written it sounds as though EIN3/EIL are the ethylene receptors.  Please rewrite to indicate that EIN3/EIL are key intermediates in ethylene signal transduction.

Lines 31-32:  Please explain better.  What are DH1588 plants and why is it significant that PsnEIL4 was only expressed at high levels in leaves of this line/

Line 32:  Please explain what “DT” is.

Lines 33-34 are hard to understand. Please rewrite for clarity and grammar.

Lines 40-46 are hard to understand. Please rewrite for clarity and grammar.

Line 54: please clarify that the ethylene receptors mark EIN2 for degradation, but do not degrade EIN2 themselves.

Lines 54-55: please clarify that EBF1/2 mark EIN3/EIL for degradation, but do not degrade EIN3/EIL themselves and that this degradation occurs in the cytoplasm.

Lines 64-66 are hard to understand. Please rewrite for clarity and grammar.

Lines 69-73 are hard to understand. Please explain better what this tells about how these different proteins interact.

Line 79: please give the full name of JA the first time it is mentioned.

Lines 81-82 are hard to understand. Please rewrite for clarity and grammar.

Line 85: please give the full names of SA and ABA the first time they are mentioned.

Lines 102-106: please break into 2 sentences and rewrite for clarity.

Section 2.2: please explain why DH poplar can thrive without any group A genes.

Lines 157-158:  I am baffled by their statement that most DH poplar EIN3/EIL genes lack 5’ and 3’ UTR.  How do they come to this conclusion? Nearly all eukaryotic genes have  5’ and 3’ UTRs!

Lines 173-175 are hard to understand. Please rewrite for clarity and grammar.

Lines 210-212 are hard to understand. Please rewrite for clarity and grammar.

Lines 224-230 are hard to understand. Please rewrite for clarity and grammar.

Lines 257-261 are hard to understand. Please rewrite for clarity and grammar, and break into at least 2 sentences.

Line 312: please state the reference gene.

Lines 335-336 should be fused into a single sentence.

Lines 372-376 are hard to understand. Please rewrite for clarity and grammar.

Comments on the Quality of English Language

The English is adequate, but there are many errors and some sentences were hard to understand.

Author Response

(The authors gave the same response as above.)

Round 2

Reviewer 1 Report

Comments and Suggestions for Authors

I thank the authors for having addressed most of my comments. A couple of requests however have not been answered so that I recall them below:

- section 2.8: in the text you state that the differences were "significant", meaning that you performed statistics on that (this applies to section 2.7 -lines 346 and 351). This part is missing in the methods, as is missing in the related Figure (Fig. 8) a tag to identify such genes. Please add related P-value as well.

A link in the text referring to Figure 8 is also missing.

Figure 8 (panels b and d): expression levels of some tissues are not visible as hidden from others (ex.: cambium and leaves of DH1588 in panel b). Error bars are still missing

Supplementary Figures are missing!

Last, since you introduce 4 mutants, it would be useful to add a Figure (perhaps you did it, but since there are no Suppl. Figures I cannot tell) and a part in the discussion where you discuss the results you obtained and whether the differences in EIL/EIN expression levels can explain the phenotypes/characteristics you observed. A brief discussion in the materials and/or introduction would be useful, but as you add a reference about their characterization, is okay

Author Response

Thank you for taking the time to review the manuscript and make valuable suggestions. I have explained it in the uploaded attachment and made detailed changes in the revised manuscript. Please review and thank you again.

Round 3

Reviewer 1 Report

Comments and Suggestions for Authors

I thank the authors for having provided an updated version of their manuscript. In my view, other than some formatting issues that will be solved on the following stages, I feel that now the manuscript can proceed for publication. My congratulations to the authors for this paper.